# Grains on the brain: A survey of dog owner purchasing habits related to grain-free dry dog foods

Sydney Banton[1], Andrew Baynham[2], Júlia G. Pezzali[1], Michael von Massow[2], Anna K. Shoveller[1] *

**1** Department of Animal Biosciences, University of Guelph, Guelph, Ontario, Canada, **2** Department of Food, Agricultural & Resource Economics, University of Guelph, Guelph, Ontario, Canada

* ashovell@uoguelph.ca

**Data Availability Statement:** All raw survey data files are available from the Agri-environmental Research Data Repository (DOI: https://doi.org/10. 5683/SP2/JZUOMW).

## Abstract

Grain-free pet food options abound in the pet food market today, representing more than 40% of available dry dog foods in the United States. There is currently a dearth of information about the factors that contribute to a dog owner's choice of a grain-free dry dog food and if those factors are similar among countries. Therefore, the primary objective of the current survey was to identify the variables that are predictive of a dog owner's choice of a grain-free dry food across North America (Canada and the United States) and Europe (France, the United Kingdom and Germany). The survey consisted of 69 questions, took less than 15 minutes to complete and was distributed virtually via Qualtrics (Qualtrics XM, Utah, USA). A total of 3,298 responses were collected, equally distributed between countries. Multinomial logistic regression was performed in SPSS Statistics (Version 26, IBM Corp, North Castle, New York, USA). Male respondents, people from France, people who ranked the importance of ingredients in a pet food in the lower quartiles and people who do not rotate their dog's diet to provide variety were less likely to select 'no grain' when choosing a pet food. In contrast, people who believe that their dog has a food allergy, follow more than 5 specific dietary routines in their own diet, do not try to include grains in their own diet, get their information about pet food from online resources or pet store staff and look for specific claims on pet food (such as 'no fillers'), were all more likely to select 'no grain' when choosing a pet food. This survey provides insight into the similarities and differences in decision making among dog owners in North America and Europe and should be considered when exploring the effects of grain-free dog foods on canine health and well-being.

## Introduction

Many dog owners today treat their dogs like family members and consequently, they consider a wide range of important variables, other than price, when selecting a diet for their dog. A survey conducted by Boya et al. [1] reported that those who anthropomorphize their dogs to a greater extent, rate price as less important when buying dog food than when buying food for

**Funding:** This research was funded by Rolf C. Hagen, Inc. (Grant 053974, website: http://caen. hagen.com) and awarded to AKS. The funders had no role in study design, data collection and analysis, or decision to publish. The funders did review the manuscript before it was submitted for publication.

**Competing interests:** I have read the journal's policy and the authors of this manuscript have the following competing interests: Sydney Banton was funded by Rolf C. Hagen, Inc. to complete the research. This does not alter our adherence to PLOS ONE policies on sharing data and materials.

their own consumption. The same survey also found that the anthropomorphization of dogs was associated with placing equal value on food quality and 'holistic', 'natural', and 'organic' claims of dog food and food for their own consumption [1].

In the human food industry, segments of the population have placed higher value on 'clean-label' foods, as they are often perceived as healthier by consumers across the globe [2]. 'Clean-label' foods often carry claims, such as, 'natural', 'organic', 'free from artificial ingredients', 'no GMOs', etc. [2]. As a result, many of these human food trends have been translated to the pet food industry, giving consumers the option to choose pet foods that carry similar claims, such as, 'no fillers', 'no artificial flavours', 'gluten-free', 'organic', and perhaps the most popular, 'grain-free'. Grain-free diets have been on the market for over a decade and make up more than 40% of dry dog foods available in the United States (USA) [3]. Dog foods that carry the claim, 'grain-free', have seen significant growth in the last decade, with sales increasing by 221% from 2012 to 2016 in the USA [4]. In a recent survey of USA pet owners, approximately 50% of respondents perceived that grain-free diets were healthier for their pet [4]. In 2015, when grain-free was nearing its peak in the USA, almost 30% of market share was composed of grain-free pet food, compared to only 15% in the United Kingdom (UK) and 1% in France in the same year [5]. This suggests that the desire for grain-free pet foods differs between countries and is valued more by North American consumers.

Although many factors, such as ingredients, price, health/nutrition and freshness are important to dog owners when choosing a dog food [1, 6, 7], all of these reports surveyed residents of the USA only. Currently, little is known about consumer habits related to the purchase of grain-free dog diets in North America and other countries around the world. In order to understand differences in consumer attitudes towards grain-free dog food, it is necessary to first understand differences in consumer demographics, diet and purchasing habits that may contribute to that. Thus, the primary objective of the current survey was to identify the factors that are predictive of a dog owner's choice of a grain-free dry dog food across Europe (France, Germany and the UK) and North America (USA and Canada). The second objective was to compare dog owner and dietary practices among countries. We hypothesized that dog owners in North America would be more likely to select grain-free compared to dog owners in Europe and that respondents who followed a more strict routine in their own diet would be more likely to select grain-free for their dog.

## Materials and methods

### Survey

The survey, titled "Pet Food Consumer Habit Survey," consisted of 69 questions, took under 15 minutes to complete and was available in English, French and German languages. The first section consisted of questions related to the respondents' demographics and diet. The diet section consisted of questions that assessed the respondent's dietary routine, if they try to eat grains as part of a healthy diet and if they try to eat more whole grains than refined grains. The second section consisted of questions related to the respondent's dog, including demographics and diet. The diet section included information on how much and how often they feed their dog via a series of true/false questions and how often they feed their dog treats, table scraps, fruits/vegetables and other items. There were also minor sections related to the dog's exercise routine and allergies, and minor sections related to the dog owner's pet food purchasing habits and factors important to dog ownership. In the allergy section, respondents were asked if their dog has ever experienced itchy skin, hair loss, smelly skin, smelly stool or soft stool and if they have ever addressed these symptoms via diet change, if they believe their dog has a food allergy, and if their dog has been diagnosed by a veterinarian with a food allergy. The purchasing

habits section consisted of questions related to where they get their information about dog food from, where they purchase their dog food from and what factors are most important to them when choosing their dog's food via a 'constant sum' question, where they were given 100 points to allocate to price, ingredient list, claims made on the bag, if your pet likes it, brand and sustainability. There were three separate questions that asked what features of pet food respondents look for when selecting a pet food. One of these questions contained 'no' statements, including 'no grain.' The term 'no grain' was intentionally used instead of the term 'grain-free' in order to avoid introducing any negative bias towards the term 'grain-free' due to the 2018 FDA report [8] suggesting a link between grain-free diets and the development of canine dilated cardiomyopathy (DCM). The last section consisted of four questions where the respondent was asked to rate how important nutrition, exercise, veterinary care and socialization are to their dog's overall health.

## Data collection

This study and survey was approved by the University of Guelph's Research Ethics Board (REB 19-12-026). Written consent was obtained from each respondent and all data was analyzed anonymously. Before completing the survey, respondents were instructed to read the "*Consent to Participate in Research*" form that outlined the purpose of the study, the researchers, procedures, risks and benefits, participation and withdrawal, their rights, confidentiality and inclusion/exclusion criteria. The survey was anonymous and no identifying information was collected.

The survey was distributed across Canada, the US, Germany, France and the UK via an online survey platform, Qualtrics (Qualtrics XM, Utah, USA). Qualtrics recruited participants via email and this was independent of the researchers. To be eligible for the survey, respondents had to be over the age of 18 years, have at least one dog that was consuming a non-prescription dry dog food (kibble) and be the primary person responsible for selecting the dog food. In addition, respondents could not be a veterinarian, dog breeder or work in the pet food industry. If the respondent had more than one dog that was consuming a non-prescription dry food, they were instructed to answer the questions according to the older dog. In order to make sure respondents were paying attention and not answering the survey questions randomly, two attention check questions were asked where the respondent was told to select a specific response. If that response was not selected, the survey terminated, and the data were not used. Those who were eligible and completed the entire survey were compensated by Qualtrics with cash, vouchers, gift-cards, or points towards rewards, etc.

In order to confirm the survey was functioning as designed, a soft launch was conducted on June 2, 2020, where 59 responses were collected from Canada, the USA and the UK, 49 responses from France and 50 responses from Germany (n = 158). The full launch occurred on June 3, 2020 and all data were collected within 24 hours of launching. The male: female quota was set to 50: 50 and the study aimed at recruiting approximately 645 responses from each country.

## Statistical analysis

All survey data were analyzed using SPSS Statistics (Version 26, IBM Corp, North Castle, New York, USA). Categorical data were analyzed from all countries together, as frequencies, and a two-sided test of equality for column proportions was done to compare proportions between countries. All pairwise comparisons were adjusted using the Bonferroni correction. Multinomial logistic regression was used to determine which variables were significant in a consumer's choice in selecting 'no grain' as a criterion for dog food. The demographic variables that were

of interest to the authors were the respondent's age, sex, country and type of dog, therefore, they were included in each model. A preliminary review of the data revealed that more than 50% of respondents in the USA were over the age of 65 years, therefore, the interaction term, USA*age 65 plus was also included in all models in order to see if the skewed age data had any effect on the outcome of the model. Age and country were input as dummy variables, where 18 to 24 years and the UK were always treated as the base case. The dependent variable was the binary outcome: selected 'no grain' or did not select 'no grain'. Statistical significance was declared at P≤0.05.

## Results

### Demographics of respondents

A total of 657 responses were collected from Canada, 657 from the USA, 656 from Germany, 661 from France and 667 from the UK (n = 3,298). Demographics of the respondents are outlined in Table 1. Results pertinent to the current discussion are outlined in this report and the complete survey can be found in Supporting Information.

The age distribution within countries was similar between France, Germany, the UK and Canada. However, in the USA more than 50% of the respondents were 65 years or older. To account for this, an interaction term of being from the USA and being 65 years or older was added to each multinomial regression model (see statistical analysis section). The sex distribution was 50% male and 50% female within each country.

### Diet of respondents

Overall, 52.4% of respondents reported that they do not follow any dietary routines themselves, while 47.6% of respondents reported that they follow at least one of the following diet regimens; low sugar or sugar-free diet (12.9%), low fat/ low calorie diet (11.5%), high protein diet (9.6%), high fiber diet (8.9%), no processed foods (i.e. no additives, minimal ingredient diet; 8.6%), organic (8.6%), low sodium diet (7.6%), vegetarian (7.1%), low or carbohydrate-free diet (5.7%), dairy-free (2.9%), diabetic diet (2.8%), gluten-free or Celiac diet (2.6%), other (2.3%), vegan (2.2%), raw (1.8%), grain-free (1.7%), Halal (1.2%), ketogenic (1.1%), Kosher (0.9%), low fiber diet (0.8%) and renal or kidney diet (0.5%). People from France (60.2%) selected none of the above more than people from Germany (40.4%) or the USA (51.9%; P<0.0001) but were not different than the UK (55.6%) or Canada (53.7%; Fig 1). In contrast, more people from Germany (7.5%) selected that they follow 5 or more dietary routines compared to France (2.1%) or the UK (1.9%; P<0.0001), but were not different from Canada (5.3%) or the USA (5.9%; Fig 1).

When asked to assign a value of 1 to 10 (0 being strongly disagree, 5 being neutral and 10 being strongly agree) to the statement, "I try to eat grains as part of a healthy diet," 68.4% of respondents assigned a value between 6–10 (considered to agree), 20.8% of respondents assigned a value of 5 (neutral), and 10.9% of respondents assigned a value between 0–4 (considered to disagree). More people from France (20.7%) disagreed with the statement than any other country (Germany: 7.2%, UK: 9.1%, Canada: 8.4%, USA: 8.8%; P<0.0001; Fig 2). Similarly, 62.6% of respondents assigned a value between 6–10 (agree) to the statement, "I try to eat more whole grains than refined grains," 19.9% of respondents assigned a 5 (neutral) and 17.5% of respondents assigned a value between 0–4 (disagree).

**Table 1. Demographic data, including sex, age, country of residence, income, education and children living in the household from all respondents (n = 3, 298).**

| Demographics | n | % |
|---|---|---|
| **Sex** | | |
| Male | 1648 | 50.0 |
| Female | 1650 | 50.0 |
| **Age** | | |
| 18–24 years | 137 | 4.2 |
| 25–34 years | 524 | 15.9 |
| 35–44 years | 680 | 20.6 |
| 45–54 years | 670 | 20.3 |
| 55–64 years | 666 | 20.2 |
| 65 years or older | 621 | 18.8 |
| **Country** | | |
| France | 661 | 20.0 |
| Germany | 656 | 19.9 |
| United Kingdom | 667 | 20.2 |
| Canada | 657 | 19.9 |
| United States | 657 | 19.9 |
| **Income** | | |
| 0–24,999 | 580 | 17.6 |
| 25,000–49,999 | 1030 | 31.2 |
| 50,000–74,999 | 671 | 20.3 |
| 75,000–99,999 | 457 | 13.9 |
| 100,000–124,999 | 216 | 6.5 |
| 125,000–149,999 | 152 | 4.6 |
| 150,000 or more | 192 | 5.8 |
| **Education** | | |
| Less than high school diploma | 215 | 6.5 |
| High school diploma | 902 | 27.3 |
| College degree | 732 | 22.2 |
| Bachelor's degree | 781 | 23.7 |
| Master's degree | 486 | 14.7 |
| Professional degree | 99 | 3.0 |
| PhD | 83 | 2.5 |
| **Live with children between 0–17 years** | | |
| Yes | 1224 | 37.1 |
| No | 2074 | 62.9 |

## Demographics of dogs

The sex, age, breed and size of the dogs owned by the respondents are reported in Table 2. A majority of respondents owned only 1 dog (78.1%), however, if the respondent owned more than one dog, they were asked to respond to the questions according to the oldest dog in their household.

  **Diet of dogs.**   When asked about what type and how frequently other foods are offered to the dog, only 2.3% of respondents reported that no other foods are offered to their dog (daily or occasionally), while 97.7% of respondents selected at least one of the following; daily dog treats (48.1%), occasional dog treats (48.3%), daily table scraps (9.3%), occasional table scraps (38.1%), daily fruits and vegetables (11.7%), occasional fruits and vegetables (26.2%), daily

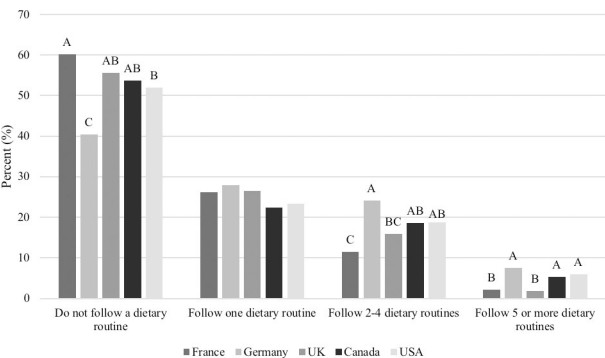

**Fig 1. Number of dietary routines followed by respondents in each country (n = 3,298).** Proportions in the same category that do not share a common letter are significantly different at P<0.05.

other foods (12.3%), and occasional other foods (13.7%). People from France were seemingly more strict with offering other foods, as 55.8% of respondents reported giving no other foods to their dog on a daily basis, which was significantly higher than the other countries (Germany: 37.3%, UK: 37.0%, Canada: 38.2%, USA: 33.3%; P<0.0001; Fig 3).

A majority of respondents answered 'true' to the following true/false questions, respectively, "I provide a fixed amount of food to my dog every day," (86.9%), "I restrict my dog's food intake to control weight," (58.4%), "I follow the feeding guidelines on my dog's pet food bag," (60.7%), "I feed my dog an amount of food based on my veterinarian's recommendation," (56.5%), and "I purposely rotate my dog's dry food to provide variety" (55.7%). More people in Europe (France: 64.4%, Germany: 64.9%, UK: 61.6%) responded that they rotate their dogs diet to provide variety compared to people in North America (Canada: 45.6%, USA: 41.6%; P<0.0001, Fig 4). Other differences among countries in their response to the above questions are outlined in Fig 4.

**Incidence of allergy symptoms in dogs.** When asked about any issues related to allergies, 40.4% of respondents reported that their dog had experienced at least one of the following; itchy skin (19.8%), soft stool (19.1%), hair loss (8.6%), smelly stool (6.7%), and smelly skin (4.9%). The number of allergy symptoms reported did not differ among countries when no symptoms, one symptom and 2 or more symptoms were compared. Of the respondents who selected one or more of the symptoms listed above, 66.4% of them reported that they have tried changing the dog's diet to address the issue. Across all respondents, only 10.9% reported

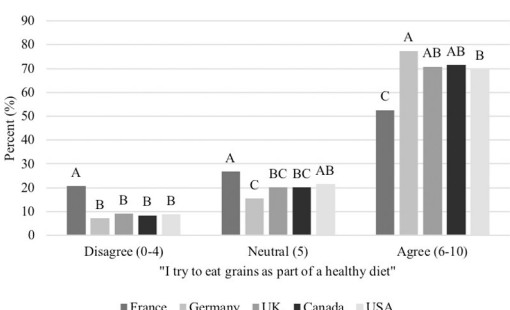

**Fig 2. Score assigned to the statement, "I try to eat grains as part of a healthy diet" between respondents in each country (n = 3,298).** Proportions in the same category that do not share a common letter are significantly different at P<0.05.

**Table 2. Demographic data, including sex, age, breed and size for all dogs (n = 3, 298).**

| Variable | n | % |
|---|---|---|
| **Sex** | | |
| Male | 1969 | 59.7 |
| Female | 1329 | 40.3 |
| **Age** | | |
| 0–2 years | 324 | 9.8 |
| 2–5 years | 1097 | 33.3 |
| 5–8 years | 892 | 27.1 |
| 8–11 years | 563 | 17.1 |
| Older than 11 years | 422 | 12.8 |
| **Breed** | | |
| Purebred | 1780 | 54.0 |
| Mixed breed | 1518 | 46.0 |
| **Size** | | |
| X-small (up to 8 lbs) | 150 | 4.5 |
| Small (8–22 lbs) | 1036 | 31.4 |
| Medium (22–55 lbs) | 1299 | 39.4 |
| Large (55–100 lbs) | 741 | 22.5 |
| Giant (over 100 lbs) | 72 | 2.2 |

feeding their dog a specific diet because they believe that their dog has a food allergy, compared to 7.8% of people who reported feeding their dog a specific diet because their dog has been diagnosed by a veterinarian with a food allergy. People from Canada (13.2%) reported feeding their dog a specific diet because they believe their dog has an allergy more often than people from the UK (8.2%; P = 0.044), but were not different from France (11.8%), Germany (9.9%) or the USA (11.0%).

**Pet food purchasing habits.** A majority of respondents selected that they get their information about pet food from their veterinarian (57.6%), followed by online resources (40.9%), pet store staff (29.3%), friends and family (27.9%), online/television advertisements (11.1%), breeders (11.0%), other resources (10.3%), animal shelters/rescues (5.5%) and from pet food companies (5.0%). More people from France (67.3%) indicated that they get their information about pet food from a veterinarian than any other country (Germany: 53.7%, UK: 48.3%, Canada: 59.7%, USA: 59.2%; P<0.0001; Fig 5). In contrast, German respondents received their information about pet food from various sources and reported getting information from online resources (55.9%) and from pet store staff (49.8%) more often than any other country (France: 29.8%, UK: 41.5%, Canada: 41.2%, USA: 35.8%; P<0.0001; France: 15.6%, UK: 30.3%, Canada: 32.4%, USA: 18.4%; P<0.0001, respectively, Fig 5). Pet food was most commonly purchased from a pet specialty store (39.9%), followed by a grocery store (31.4%), online (16.3%), other (8.1%), and veterinary clinics (4.3%).

There were three separate questions that assessed what attributes respondents consider when choosing a pet food. Claims related to targeted nutrition resonated with a majority of respondents (84.5%) who selected at least one of the following options: 'age specific nutrition' (47.6%), 'size specific nutrition' (47.2%), 'digestive care' (26.9%), 'dental care' (22.3%), 'joint care' (20.7%), 'breed specific nutrition' (19.1%), 'weight control' (17.5%) and 'sensitive skin/ stomach' (10.7%). More respondents in France (91.7%) selected at least one of the above options compared to respondents in the UK (83.4%), Canada (83.9%) or the USA (75.5%; P<0.0001; Fig 6) but did not differ from Germany (88.0%).

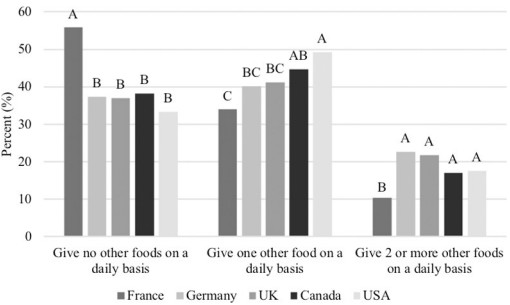

**Fig 3. Amount of other foods (respondents had the options of dog treats, table scraps, fruits/vegetables and other foods) given on a daily basis among respondents in each country (n = 3,298).** Proportions in the same category that do not share a common letter are significantly different at P<0.05.

With respect to protein related claims, 89.8% of respondents selected that they look for at least one of the following options: 'poultry' (68.5%), 'beef' (55.8%), 'fish' (32.6%), 'protein #1 ingredient' (24.1%), 'organic/natural' (21.8%), 'pork' (21.2%), 'limited ingredient diet' (13.6%), 'exotic proteins' (9.7%) and 'vegetarian/vegan diet' (3.2%). Similar to the above question, more people in France (92.7%) selected at least one of the above options compared to people in Germany (88.1%) or the USA (86.3%; P<0.0001; Fig 6) but did not differ among respondents in the UK (92.1%) or Canada (89.5%).

Compared to the attributes listed above, the claims containing 'no' statements held the lowest value among respondents, but still a majority (61.8%) of respondents selected at least one of the following options: 'no artificial colours or flavours' (38.6%), 'no fillers' (30.1%), 'no by-products' (28.3%), 'no grain' (21.3%), 'no corn' (16%), 'no wheat' (15.7%), 'no soy' (14.5%), 'no pulse ingredients' (9.1%) and 'gluten-free' (7.7%). In contrast to the above questions, fewer respondents in France (44.9%) selected at least one of 'no' statements compared to the other four countries (P<0.0001; Fig 6). However, there was no difference among Germany (68.8%), Canada (69.5%) or the USA (65.8%; Fig 6). When just the claim, 'no grain', was compared between countries, again, respondents from France (8.0%) demonstrated the lowest tendency to select 'no grain' as an attribute that influences the purchase of pet food compared to the other countries (Germany, 30.0%, UK, 19.6%, Canada, 21.8%, USA, 27.4%; P<0.0001, Fig 7).

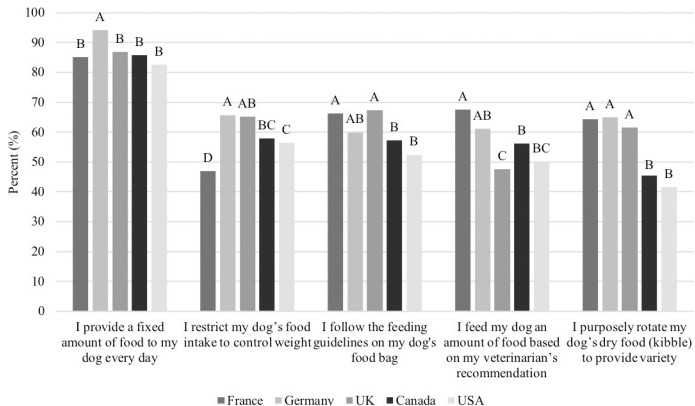

**Fig 4. Percent of respondents that answered 'true' to the five feeding statements, among countries (n = 3,298).** Proportions in the same category that do not share a common letter are significantly different at P<0.05.

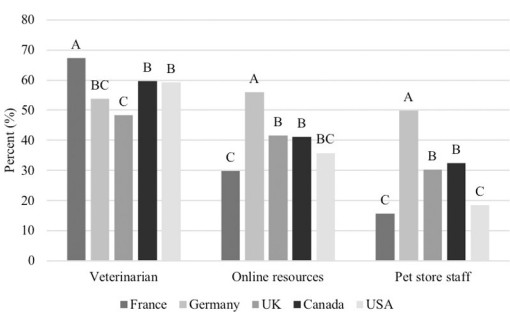

**Fig 5. Percent of respondents that get their information about pet food from a veterinarian, online and/or from pet store staff among countries (n = 3,298).** Proportions in the same category that do not share a common letter are significantly different at P<0.05.

When asked to allocate 100 points between six attributes of pet food, in order of importance, owners indicated that the perception that their dog enjoys consuming the food was most important (29.4 ± 21.4 points), followed by the ingredient list (26.1 ± 21.4 points), the price (19.5 ± 18.9 points), the brand of dog food (11.6 ± 14.8 points), claims made on the bag (7.3 ± 9.5 points) and sustainability (6.1 ± 8.7 points). When country-specific rankings were evaluated, respondents in France (22.7 ± 0.8 points) allocated more points to the price of the dog food compared to Germany (17.4 ± 0.7 points), the USA (17.5 ± 0.8 points) or the UK (17.4 ± 0.7 points; P<0.05), while respondents in Germany (28.9 ± 0.8 points), Canada (28.6 ± 0.9 points) and the USA (29.0 ± 1.0 points) allocated more points to the ingredient list compared to France (21.5 ± 0.7 points) or the UK (22.7 ± 0.7 points; P<0.05; Fig 8). Other differences among countries in importance of specific attributes are outlined in Fig 8.

**Factors important to dog ownership.** When asked to rank the following items on a scale of 0 to 10, 97% of respondents indicated that nutrition was an important component of dog ownership (assigned a score between 6 to 10), with 40.7% of them indicating that nutrition was extremely important (score of 10). Similar scores were assigned to exercise and regular veterinary care/vaccines, where 94.7% and 90.2% of respondents assigned a score between 6 to 10, respectively, and 39.8% and 44.3% of respondents assigned a score of 10, respectively. Finally, although scores were still high, socialization was shown to be the least important factor for dog ownership. Across all respondents, 85.8% assigned a score between 6 to 10, with 28.9%

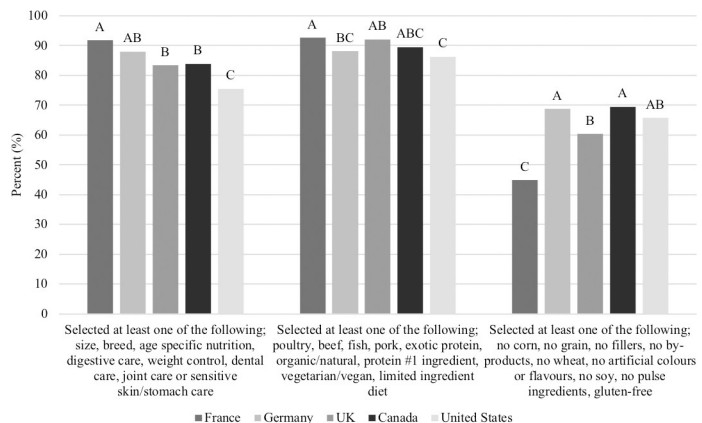

**Fig 6. Percent of respondents that look for at least one of the above options when selecting a pet food (n = 3,298).** Proportions in the same category that do not share a common letter are significantly different at P<0.05.

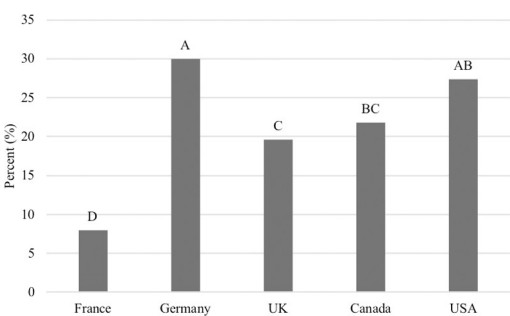

**Fig 7. Percent of respondents in each country that look for "no grain" when selecting a pet food (n = 3,298).**
Proportions that do not share a common letter are significantly different at P<0.05.

of them selecting 10, indicating that socialization is extremely important. While all five countries assigned similar scores, when compared, there were some differences, outlined in Fig 9.

**Multinomial logistic regression.** The models presented below were developed with the goal of understanding what variables are predictive of a consumer's choice of a grain-free dog diet. Seven models were developed using questions from the survey related to allergies in dogs, human and dog diet, pet food purchasing habits, veterinary care/advice, dog ownership, demographics and exercise/feeding practices. The models with a McFadden Pseudo R-Square value above 0.10 are discussed in detail and results are reported in tables. The other models are discussed briefly and tables can be found in Supplementary Data.

The Allergy Model ($\chi^2$(23) = 383.24, P<0.0001, Table 3) had a McFadden Pseudo R-Square value of 0.11. Male respondents were less likely to select 'no grain' than female respondents (P = 0.015). People from France were less likely to select 'no grain' than people from the UK (P<0.0001). People who selected 'yes' to feeding their dog a specific diet because it has been diagnosed by a veterinarian with a food allergy were less likely to select 'no grain' than those who selected 'no' (P = 0.041). In contrast, people who selected that their dog has experienced two or more allergy symptoms were 1.3 times more likely to select 'no grain' than those who selected no allergy symptoms (P = 0.047). People who selected 'yes' to feeding their dog a specific diet because they believe it has a food allergy were 4 times more likely to select 'no grain' than those who selected 'no' (P<0.0001). When each country was looked at separately for this question, people from Germany and France were not significantly different from UK. People

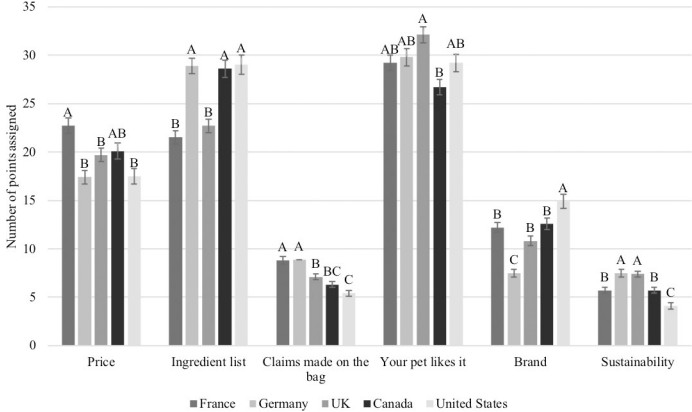

**Fig 8. Mean (±SD) score assigned to each factor involved in selecting a pet food among countries (n = 3,298).**
Proportions in the same category that do not share a common letter are significantly different at P<0.05.

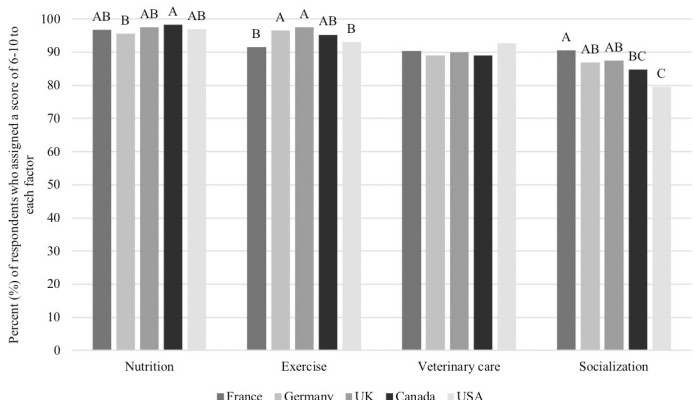

**Fig 9. Percent of respondents who assigned a value of 6–10 to the factors important to dog ownership among countries (n = 3,298).** Proportions in the same category that do not share a common letter are significantly different at P<0.05.

from the USA were still more likely to select 'no grain' if they said 'yes' to feeding a specific diet because they believe their dog has an allergy, but significantly less likely than those from the UK (calculated Odds Ratio: 1.79, P = 0.047). People who selected that they look for claims that are often linked to allergies in dogs: sensitive skin/stomach formulas, exotic proteins or limited ingredient diets when choosing a pet food were 1.7, 1.9 and 3 times more likely, respectively, to select 'no grain' than those who do not look for these options (P<0.0001). Age, type of dog and USA by 65 years plus interaction was not significant (P>0.05).

The Diet Model ($\chi^2$(29) = 587.81, P<0.0001, Table 4) had a McFadden Pseudo R-Square value of 0.17. Male respondents were less likely to select 'no grain' than female respondents (P = 0.046). People from France (P<0.0001) and people from Canada (P = 0.024) were less likely to select 'no grain' than people from the UK. When allotting points to importance of the ingredient list on a pet food label (more points being more important), people that were in the first, second and third quartiles were less likely to select 'no grain' than those in the fourth quartile (P<0.0001). People from Germany were 1.4 times more likely to select 'no grain' than those from the UK (P = 0.019). People who selected that they follow 5 or more dietary routines for themselves were 2.3 times more likely to select 'no grain' than those that do not follow any dietary routines (P<0.0001). People who selected between 0–4 (disagree) or 5 (neutral) in response to, "I try to eat grains as part of a healthy diet" were 1.6 and 1.4 times more likely to select 'no grain' than those that selected between 6–10 (P = 0.003). People who selected between 1–3 or 4 or more options in the protein ingredient pet food question were 1.6 and 1.8 times more likely to select 'no grain' than those who selected no options (P = 0.019 and P = 0.010, respectively). People who selected that they look for 'no fillers' or 'no by-products' when choosing a pet food were 2.6 and 1.6 times more likely to select 'no grain,' respectively, than those who do not look for 'no fillers' or 'no by-products' (P<0.0001). People who give their dog 2 or more different foods (treats, table scraps, fruits/veggies, other foods) on a daily basis were 1.4 times more likely to select 'no grain' than those that do not give their dog any other foods on a daily basis (P = 0.006). Age, type of dog, number of choices selected in the targeted nutrition pet food question, importance of nutrition and the USA by 65 plus interaction were not significant (P>0.05). In addition, when the 20 human diet options were the only variables in the model, only those that selected that they follow a grain-free, vegetarian, vegan, ketogenic or no processed foods diet were more likely to select 'no grain' when choosing a pet food (P<0.05).

**Table 3. Multinomial logistic regression parameter estimates for the Allergy Model.**

| Variable | β[1] | Std. Error | P-Value | OR[2] | 95% CI[3] Lower Bound | 95% CI[3] Upper Bound |
|---|---|---|---|---|---|---|
| **Age** | | | | | | |
| 25 to 34 years | -0.040 | 0.261 | 0.880 | 0.961 | 0.576 | 1.605 |
| 35 to 44 years | -0.100 | 0.256 | 0.694 | 0.904 | 0.548 | 1.493 |
| 45 to 54 years | 0.128 | 0.255 | 0.616 | 1.137 | 0.689 | 1.875 |
| 55 to 64 years | 0.181 | 0.256 | 0.479 | 1.199 | 0.726 | 1.979 |
| 65 years or older | 0.178 | 0.282 | 0.528 | 1.195 | 0.687 | 2.078 |
| 18 to 24 years | . | . | . | . | . | . |
| **Sex** | | | | | | |
| Male | -0.224 | 0.092 | 0.015 | 0.799 | 0.667 | 0.958 |
| Female | . | . | . | . | . | . |
| **Country** | | | | | | |
| Germany | 0.281 | 0.150 | 0.062 | 1.324 | 0.986 | 1.778 |
| France | -1.042 | 0.199 | <0.0001 | 0.353 | 0.239 | 0.521 |
| USA | 0.113 | 0.189 | 0.550 | 1.119 | 0.773 | 1.620 |
| Canada | 0.016 | 0.157 | 0.916 | 1.017 | 0.748 | 1.382 |
| UK | . | . | . | . | . | . |
| **Type of dog** | | | | | | |
| Purebred | -0.026 | 0.092 | 0.780 | 0.975 | 0.813 | 1.168 |
| Mixed breed | . | . | . | . | . | . |
| **Allergy Symptoms** | | | | | | |
| One | 0.090 | 0.112 | 0.422 | 1.094 | 0.879 | 1.362 |
| Two or more | 0.278 | 0.140 | 0.047 | 1.321 | 1.003 | 1.740 |
| None | . | . | . | . | . | . |
| **Do you feed your dog a specific diet because you believe your dog has a food allergy?** | | | | | | |
| Yes | 1.389 | 0.327 | <0.0001 | 4.010 | 2.110 | 7.618 |
| No | . | . | . | . | . | . |
| **Do you feed your dog a specific diet because your dog has been diagnosed by a veterinarian with a food allergy?** | | | | | | |
| Yes | -0.458 | 0.224 | 0.041 | 0.633 | 0.408 | 0.982 |
| No | . | . | . | . | . | . |
| **When choosing a pet food, I look for . . . Sensitive skin/stomach** | 0.526 | 0.139 | <0.0001 | 1.692 | 1.288 | 2.222 |
| **When choosing a pet food, I look for . . . Limited ingredient diet** | 1.105 | 0.117 | <0.0001 | 3.019 | 2.401 | 3.795 |
| **When choosing a pet food, I look for . . . Exotic protein** | 0.644 | 0.138 | <0.0001 | 1.904 | 1.452 | 2.496 |
| **Age 65 plus\*USA** | 0.425 | 0.253 | 0.093 | - | - | - |
| **Germany\*Do you feed your dog a specific diet because you believe your dog has a food allergy?** | -0.101 | 0.425 | 0.813 | - | - | - |
| **France\*Do you feed your dog a specific diet because you believe your dog has a food allergy?** | -0.320 | 0.466 | 0.493 | - | - | - |
| **USA\*Do you feed your dog a specific diet because you believe your dog has a food allergy?** | -0.805 | 0.406 | 0.047 | - | - | - |
| **Canada\*Do you feed your dog a specific diet because you believe your dog has a food allergy?** | -0.730 | 0.401 | 0.069 | - | - | - |

[1] Estimated multinomial logistic regression coefficient

[2] Odds Ratio or exponentiation of the coefficient (β)

[3] 95% Confidence Interval of the Odds Ratio

McFadden Pseudo R-Square = 0.11

Dependent variable categories, 1 = selected 'no grain', 0 = did not select 'no grain'

**Table 4. Multinomial logistic regression parameter estimates for the Diet Model.**

| Variable | β[1] | Std. Error | P-Value | OR[2] | 95% CI[3] Lower Bound | Upper Bound |
|---|---|---|---|---|---|---|
| **Age** | | | | | | |
| 25 to 34 years | 0.032 | 0.267 | 0.904 | 1.033 | 0.612 | 1.743 |
| 35 to 44 years | -0.181 | 0.263 | 0.490 | 0.834 | 0.499 | 1.396 |
| 45 to 54 years | -0.049 | 0.262 | 0.851 | 0.952 | 0.570 | 1.590 |
| 55 to 64 years | -0.050 | 0.264 | 0.850 | 0.951 | 0.567 | 1.595 |
| 65 years or older | -0.077 | 0.290 | 0.790 | 0.926 | 0.524 | 1.634 |
| 18 to 24 years | . | . | . | . | . | . |
| **Sex** | | | | | | |
| Male | -0.191 | 0.096 | 0.046 | 0.826 | 0.685 | 0.997 |
| Female | . | . | . | . | . | . |
| **Country** | | | | | | |
| Germany | 0.337 | 0.144 | 0.019 | 1.401 | 1.057 | 1.856 |
| France | -0.976 | 0.187 | <0.0001 | 0.377 | 0.261 | 0.543 |
| USA | -0.148 | 0.189 | 0.433 | 0.862 | 0.595 | 1.249 |
| Canada | -0.344 | 0.152 | 0.024 | 0.709 | 0.526 | 0.955 |
| UK | . | . | . | . | . | . |
| **Type of dog** | | | | | | |
| Purebred | 0.005 | 0.096 | 0.958 | 1.005 | 0.833 | 1.213 |
| Mixed breed | . | . | . | . | . | . |
| **Which one of the following best reflects your own dietary routine (select all that apply)** | | | | | | |
| Selected 1 option | -0.021 | 0.119 | 0.859 | 0.979 | 0.775 | 1.237 |
| Selected 2–4 options | 0.131 | 0.127 | 0.304 | 1.140 | 0.888 | 1.462 |
| Selected 5 or more options | 0.825 | 0.202 | <0.0001 | 2.282 | 1.535 | 3.394 |
| Selected no options | . | . | . | . | . | . |
| **I try to eat grains as part of a healthy diet** | | | | | | |
| 0–4 (Disagree) | 0.498 | 0.169 | 0.003 | 1.646 | 1.183 | 2.290 |
| 5 (Neutral) | 0.364 | 0.124 | 0.003 | 1.439 | 1.129 | 1.835 |
| 6–10 (Agree) | . | . | . | . | . | . |
| **When choosing a pet food, I look for (size, breed, age specific nutrition, digestive, weight, dental, joint, sensitive skin/stomach care)** | | | | | | |
| 1–3 options selected | 0.054 | 0.149 | 0.718 | 1.056 | 0.788 | 1.415 |
| 4 or more options selected | 0.212 | 0.176 | 0.227 | 1.236 | 0.876 | 1.744 |
| No options selected | . | . | . | . | . | . |
| **When choosing a pet food, I look for (poultry, beef, fish, pork, exotic protein, organic/natural, protein #1 ingredient, vegetarian/vegan, limited ingredient diet)** | | | | | | |
| 1–3 options selected | 0.485 | 0.207 | 0.019 | 1.625 | 1.082 | 2.440 |
| 4 or more options selected | 0.565 | 0.220 | 0.010 | 1.759 | 1.143 | 2.707 |
| No options selected | . | . | . | . | . | . |
| **When choosing a pet food, I look for . . .** | | | | | | |
| No fillers | 0.964 | 0.111 | <0.0001 | 2.621 | 2.108 | 3.260 |
| No by-products | 0.440 | 0.111 | <0.0001 | 1.553 | 1.248 | 1.932 |
| **Other food items (dog treats, table scraps, fruits/veggies, other) given on a daily basis** | | | | | | |
| One option selected | 0.054 | 0.109 | 0.619 | 1.056 | 0.853 | 1.307 |
| Two or more options selected | 0.364 | 0.131 | 0.006 | 1.439 | 1.113 | 1.862 |
| No options selected | . | . | . | . | . | . |

(*Continued*)

**Table 4.** (Continued)

| Variable | $\beta$[1] | Std. Error | P-Value | OR[2] | 95% CI[3] | |
|---|---|---|---|---|---|---|
| | | | | | Lower Bound | Upper Bound |
| **How important are the following items when purchasing pet food . . .** | | | | | | |
| **Ingredients** | | | | | | |
| First Quartile | -1.326 | 0.150 | <0.0001 | 0.266 | 0.198 | 0.357 |
| Second Quartile | -0.827 | 0.136 | <0.0001 | 0.437 | 0.335 | 0.571 |
| Third Quartile | -0.433 | 0.124 | <0.0001 | 0.649 | 0.509 | 0.827 |
| Fourth Quartile | . | . | . | . | . | . |
| **Importance of nutrition in terms of your dog's overall health** | | | | | | |
| 2–5 (Not important/neutral) | 0.273 | 0.348 | 0.433 | 1.314 | 0.664 | 2.598 |
| 6–10 (Important) | . | . | . | . | . | . |
| **Age 65 plus*USA** | 0.300 | 0.266 | 0.259 | - | - | - |

[1] Estimated multinomial logistic regression coefficient

[2] Odds Ratio or exponentiation of the coefficient (β)

[3] 95% Confidence Interval of the Odds Ratio

McFadden Pseudo R-Square = 0.17

Dependent variable categories, 1 = selected 'no grain', 0 = did not select 'no grain'

The Purchasing Habits Model ($\chi^2(26)$ = 438.75, P<0.0001, Table 5) had a McFadden Pseudo R-Square value of 0.13. Again, people from France were less likely to select 'no grain' than people from the UK (P<0.0001) and male respondents were less likely to select 'no grain' than female respondents (P = 0.002). People who purposely rotate their dog's dry food to provide variety were less likely to select 'no grain' (P = 0.021). Similar to the Diet Model, people who ranked the importance of ingredients on a pet food label in the first, second and third quartiles were less likely to select 'no grain' than those who ranked ingredients in the fourth quartile (P<0.0001). People who selected that they get their information about pet food from online resources and/or pet store staff were 1.6 and 1.3 times more likely to select 'no grain' than those that do not get their information about pet food online or from pet store staff (P<0.0001 and P = 0.006, respectively). People who purchase their dog food from a veterinary clinic, a pet specialty store and online were 2.2, 1.6 and 2.4 times more likely to select 'no grain' than people who purchase their pet food from a grocery store (P = 0.002, P<0.0001 and P<0.0001, respectively). Age, type of dog, importance of price when selecting pet food and the interaction term, USA by 65 plus were not significant (P>0.05).

The other models all had a McFadden Pseudo R-Square value less than 0.10 (The Vet Model: $\chi^2(21)$ = 169.02, P<0.0001, McFadden Pseudo R-Square value of 0.049, The Dog Ownership Model: $\chi^2(29)$ = 226.90, P<0.0001, McFadden Pseudo R-Square value of 0.066, The Demographics Model: $\chi^2(33)$ = 208.24, P<0.0001, McFadden Pseudo R-Square value of 0.061, the Exercise/Feeding Model: $\chi^2(29)$ = 204.07, P<0.0001, McFadden Pseudo R-Square value of 0.060, S1–S4 Tables). In general, across these models, male respondents were less likely to select 'no grain' than female respondents, people from France were less likely to select 'no grain' than people from the UK and people from Germany were more likely to select 'no grain' than people from the UK (P<0.05). Interestingly, from the Vet Model, only getting your information about pet food from the vet was not significant and from the Demographics Model, education level and income were also not significant (P>0.05).

**Table 5. Multinomial logistic regression parameter estimates for the Purchasing Habits Model.**

| Variable | β[1] | Std. Error | P-Value | OR[2] | 95% CI[3] | |
| --- | --- | --- | --- | --- | --- | --- |
| | | | | | Lower Bound | Upper Bound |
| **Age** | | | | | | |
| 25 to 34 years | 0.039 | 0.263 | 0.883 | 1.039 | 0.621 | 1.739 |
| 35 to 44 years | -0.148 | 0.258 | 0.565 | 0.862 | 0.520 | 1.429 |
| 45 to 54 years | 0.084 | 0.257 | 0.745 | 1.087 | 0.657 | 1.799 |
| 55 to 64 years | 0.160 | 0.259 | 0.537 | 1.174 | 0.706 | 1.952 |
| 65 years or older | 0.151 | 0.287 | 0.598 | 1.163 | 0.663 | 2.041 |
| 18 to 24 years | . | . | . | . | . | . |
| **Sex** | | | | | | |
| Male | -0.283 | 0.093 | 0.002 | 0.753 | 0.628 | 0.903 |
| Female | . | . | . | . | . | . |
| **Country** | | | | | | |
| Germany | 0.214 | 0.141 | 0.130 | 1.239 | 0.939 | 1.633 |
| France | -1.009 | 0.185 | <0.0001 | 0.365 | 0.254 | 0.524 |
| USA | 0.051 | 0.182 | 0.779 | 1.052 | 0.736 | 1.505 |
| Canada | -0.044 | 0.149 | 0.768 | 0.957 | 0.714 | 1.282 |
| UK | . | . | . | . | . | . |
| **Type of Dog** | | | | | | |
| Purebred | -0.061 | 0.093 | 0.514 | 0.941 | 0.784 | 1.129 |
| Mixed breed | . | . | . | . | . | . |
| **I purposely rotate my dog's dry food to provide variety** | | | | | | |
| True | -0.224 | 0.097 | 0.021 | 0.799 | 0.661 | 0.967 |
| False | . | . | . | . | . | . |
| **Where do you get your information about dog food from?** | | | | | | |
| Veterinarian | -0.116 | 0.097 | 0.232 | 0.891 | 0.737 | 1.077 |
| Online | 0.452 | 0.096 | <0.0001 | 1.571 | 1.301 | 1.897 |
| Pet store staff | 0.290 | 0.105 | 0.006 | 1.337 | 1.087 | 1.643 |
| **Where do you purchase your pet food from?** | | | | | | |
| Vet clinic | 0.787 | 0.250 | 0.002 | 2.197 | 1.347 | 3.583 |
| Pet specialty store | 0.452 | 0.123 | <0.0001 | 1.572 | 1.235 | 2.001 |
| Online | 0.863 | 0.144 | <0.0001 | 2.371 | 1.789 | 3.143 |
| Other | 0.304 | 0.215 | 0.156 | 1.355 | 0.890 | 2.064 |
| Grocery store | . | . | . | . | . | . |
| **How important are the following items when purchasing pet food . . . Price** | | | | | | |
| First Quartile | 0.024 | 0.145 | 0.866 | 1.025 | 0.772 | 1.361 |
| Second Quartile | 0.167 | 0.139 | 0.228 | 1.182 | 0.901 | 1.551 |
| Third Quartile | 0.090 | 0.144 | 0.533 | 1.094 | 0.825 | 1.450 |
| Fourth Quartile | . | . | . | . | . | . |
| **How important are the following items when purchasing pet food . . . Ingredients** | | | | | | |
| First Quartile | -1.552 | 0.147 | <0.0001 | 0.212 | 0.159 | 0.283 |
| Second Quartile | -0.876 | 0.138 | <0.0001 | 0.416 | 0.318 | 0.545 |
| Third Quartile | -0.439 | 0.123 | <0.0001 | 0.645 | 0.507 | 0.820 |
| Fourth Quartile | . | . | . | . | . | . |

(*Continued*)

**Table 5.** (Continued)

| Variable | β[1] | Std. Error | P-Value | OR[2] | 95% CI[3] | |
|---|---|---|---|---|---|---|
| | | | | | Lower Bound | Upper Bound |
| **Age 65 plus*USA** | 0.370 | 0.258 | 0.151 | - | - | - |

[1] Estimated multinomial logistic regression coefficient

[2] Odds Ratio or exponentiation of the coefficient (β)

[3] 95% Confidence Interval of the Odds Ratio

McFadden Pseudo R-Square = 0.13

Dependent variable categories, 1 = selected 'no grain', 0 = did not select 'no grain'

## Discussion

To the authors' knowledge, this is the first large scale, published survey investigating consumer habits related to purchasing grain-free dry dog food across the UK, Germany, France, the USA and Canada. In addition, this survey was entirely recruited via Qualtrics, therefore there was no bias in social media recruitment or location bias. As expected, many factors contribute to a dog owner's choice of grain-free dry dog food. Perhaps the most noteworthy from this study, the incidence of allergies, owner purchasing habits and the dietary routine of owners appear to have the highest explanatory power for choice of grain-free dry dog food, according to results from multinomial logistic regression.

From the Allergy Model, this study found that people who reported that their dog experienced two or more allergy symptoms were more likely to select 'no grain' than those that reported no symptoms. The most common food allergens in dogs with diagnosed food allergies are beef and dairy, accounting for more than 60% of food allergy cases reviewed by Verlinden et al. [9], followed by wheat, egg and chicken, while the least common food allergens in dogs are soy, corn, fish, pork, and rice (each accounting for less than 6% of all food allergies diagnosed) [10]. While animal protein allergies are far more common than grain allergies in dogs [9, 10], the results from the current study suggest that consumers may believe that a grain-free diet can address allergy symptoms in their dog. In addition, selecting a claim that is often advertised as being beneficial to the treatment of allergies in dogs, such as, 'limited ingredient diet', 'sensitive skin/stomach' and 'exotic protein' was also positively associated with selecting a grain-free diet.

Another finding in the present study was that people who believe their dog has a food allergy were four times more likely to select 'no grain' when choosing a pet food. In contrast, owners whose dog has been diagnosed by a veterinarian with a food allergy were less likely to select 'no grain'. Veterinarians are trained to diagnose a food allergy via an elimination trial where a hydrolyzed protein diet is fed exclusively for 6–8 weeks. However, this can be difficult for a dog owner to comply with since only one specific diet and no other foods can be fed for the duration of the trial. Therefore, it is possible that dog owners are turning to grain-free as their own remedy to a suspected food allergy. One flaw in interpreting this allergy data is the fact that the survey excluded people feeding a prescription diet to their dog and likely, if a dog is being fed a specific diet because their vet diagnosed an allergy, it would be a prescription diet. Another possibility is that, although 7.8% of people reported that they feed their dog a specific diet because their vet diagnosed a food allergy, these may be prescription diets, as we did not define what a prescription diet was to the respondent or ask what diet they are currently feeding, in order to confirm.

We hypothesized that those who follow a stricter dietary routine themselves would be more likely to impose dietary restrictions on their dogs and select a grain-free diet, since grain-free

diets are often perceived as healthier by pet food consumers [4]. Modelling data from Kumcu & Woolverton [11] demonstrated that people who purchase 'premium' food for themselves (defined as USDA certified organic food) are more likely to purchase 'premium' pet food (defined according to brand and keywords such as natural, organic, etc). In a survey by Dodd et al. [12], people who reported feeding their dog a plant-based diet were all either vegan or vegetarian themselves. These results align with our findings where people who follow 5 or more dietary routines themselves and people who do not try to eat grains as part of a healthy diet were more likely to select 'no grain' when choosing a pet food. Furthermore, those who follow a grain-free, vegetarian, vegan, ketogenic or no processed foods dietary routine themselves were all more likely to select 'no grain' when choosing a pet food. It is unclear why people who are vegetarian or vegan would be more likely to select a grain free diet for their dog, however, this highlights the idea that many variables, not only owner dietary habits, contribute to selection of a dog food. It is necessary to point out that the current survey only included those feeding a dry food (kibble) diet and these data may be quite different if owners who feed a different dietary format are included.

People who allocate more points to the importance of the ingredients in a pet food and people who look for the terms, 'no fillers' and 'no by-products' were more likely to select 'no grain' when choosing a pet food, supporting the statement by LaFlamme et al. [13] that grains are often considered 'fillers' by pet owners. In addition, Conway & Saker [14] reported that respondents in a survey often selected that grain-free diets were "diets free of fillers and by-products". This is not surprising, given that ingredients in pet food are consistently ranked within the top three most important factors by consumers [7, 14] and were ranked as the second most important factor in the current survey. In addition, respondents were willing to pay more for natural and organic ingredients compared to conventional ingredients in a discrete choice experiment conducted by Simonsen et al. [6]. This suggests that those who value organic and natural ingredients share a similar mind set with those who select grain-free diets. Furthermore, this suggests that those who believe ingredients in a dog food are more important than price may be more susceptible to advertising claims on dog foods, such as 'no fillers' and 'no by-products'. Although it seems that ingredients are one of the most important factors driving pet food purchasing choices, animals require nutrients, not ingredients [15]. Taking the above findings together, it is possible that dog owners are using ingredients as a proxy for nutrition and if they perceive the ingredients as healthy, they are more likely to feed it to their dog. This may also help explain the finding that those who selected that they give two or more other foods on a daily basis were more likely to select 'no grain' when choosing a pet food. It is possible that these people are supplementing their dog's diet with other foods they perceive as healthy, similar to how those that feed a raw diet will often supplement with oils, fruits, vegetables, etc. [16]. However, since we did not ask more specific questions about adding additional foods to a dry dog food, we cannot draw any concrete conclusions related to the potential effect of these additional foods.

The above conclusions highlight an important disconnect between human and pet nutrition. While the majority of European and North American food-based dietary guidelines recommend consuming whole grains as part of a healthy diet [17], grains appear to be perceived as unhealthy in a dog's diet by pet owners. There is ample evidence that consuming whole grains and fibre can lower human risk for cardiovascular disease [18] and no scientific evidence that grains are detrimental to a dog's health. This suggests that the beliefs dog owners have about grains may be largely based on marketing strategies, which is consistent with the findings that selecting claims, 'no fillers' and 'no by-products' are associated with selecting 'no grain.' However, allowing consumers to drive ingredient composition of pet food could potentially lead to nutrient imbalances if amino acid balance is not considered [15, 19]. Recently,

regular consumption of a grain-free diet has been associated with the development of canine dilated cardiomyopathy (DCM) [8]. Grain-free diets replace grain ingredients with pulse ingredients. Although high in protein, the amino acid profile of a pulse is different than that of a grain. Thus, careful consideration should be taken with regards to the amino acid composition and balance of the diet when grains are completely replaced by legumes and/or pulses. Furthermore, the Purchasing Habits Model found that those who rotate their dogs dry food to provide variety are less likely to select 'no grain', suggesting that grain-free diets may be selected as a static and regimented dietary routine. A recent clinical study by Kaplan et al. [20] revealed that in 24 Golden Retrievers diagnosed with DCM, the median length of time consuming the same grain-fee diet was 814 days, and the maximum, 3,558 days. In human nutrition, a diverse diet is encouraged to ensure breadth of nutrient intake to maintain health. It has been well established that those in developing countries, particularly women, who lack a diverse diet are more likely to develop micronutrient deficiencies [21]. Furthermore, using data from more than 1,000 French adults, Bianchi et al. [22] showed that adults with a higher diet diversity score were more likely to have a higher nutritional adequacy score. Although the importance of a diverse diet is established in human nutrition, there is a dearth of data with regards to the metabolic and health impact of a static feeding regimen in the domestic dog. Thus, future studies should investigate the effects of static feeding of a grain-free diet on micronutrient imbalances and its impact on the development of nutritionally-mediated DCM.

Evaluation of consumer purchasing behaviours across countries and between sexes is a strength of this study. Across all models, men were less likely to select 'no grain' when choosing a pet food. Several human food purchasing studies around the world have found that women are more likely to purchase organic foods than men [23–25], thus it was expected that this trend would translate to purchasing pet food. Similarly, respondents from France were always less likely to select 'no grain'. More than half of the respondents from France did not follow any dietary routine themselves, they were less likely to get information about pet food from online resources and pet store staff and they rated the ingredient list on pet foods as less important, all of which were factors that were inversely associated with selecting a grain-free dog food. In contrast, people from France rated price as more important and selected targeted nutrition options (i.e.–size, breed, age specific nutrition, etc.) more often, suggesting that other factors may be more important to them than whether or not the diet contains grains.

Divergence in opinions about grain-free dog foods in Europe are mirrored in sales data. Although four of the top ten European pet food companies have plants that are located in Germany or France and produce grain-free diet options [26], the number of pet foods with the claims 'no additives/preservatives' and 'all natural product' were 4 and 7 times higher, respectively, in Germany compared to France from 2015 to 2018 [27]. In this survey, respondents from Germany were more likely to select 'no grain' in four of seven regression models. In terms of human food, France and Germany are the top two largest markets for organic food consumption, following the USA [28], so perhaps the desire to impart your own dietary choices on your dog is not the same among countries. Further to that idea, more respondents in France disagreed with the statement, "I try to eat grains as part of a healthy diet" compared to any other country, despite the food-based dietary guidelines in France recommending consumption of starchy foods, especially whole grains, at each meal [29]. This implies that people from France do not impart their own dietary choices on their pets, compared to the other countries. In addition, it is possible that countries in Europe view their pet dogs differently, however, the human-animal bond in different countries around the world needs further investigation.

## Conclusion

Overall, while it is clear that many factors contribute to a consumer selecting a grain-free dry dog food, in the current study, some factors such as: sex, perception of allergies, a consumer's own dietary routine, different pet food information resources, the importance of the ingredient list and looking for specific claims, may be more predictive than others. Furthermore, the choice of grain-free is not the same between continents or even between countries, as people from France were always less likely to select 'no grain', while people from Germany were often more likely to select 'no grain'. These factors should be considered when exploring the population level effects of grain-free dog foods on canine health and well-being. In addition, the data from the present survey lends insights into effective avenues for the scientific community to facilitate knowledge transfer between themselves and consumers.

## Supporting information

**S1 Table. Multinomial logistic regression parameter estimates for the Vet Model.** [1] Estimated multinomial logistic regression coefficient. [2] Odds Ratio or exponentiation of the coefficient (β). [3] 95% Confidence Interval of the Odds Ratio. McFadden Pseudo R-Square = 0.050. Dependent variable categories, 1 = selected 'no grain', 0 = did not select 'no grain'.
(DOCX)

**S2 Table. Multinomial logistic regression parameter estimates for the Dog Ownership Model.** [1] Estimated multinomial logistic regression coefficient. [2] Odds Ratio or exponentiation of the coefficient (β). [3] 95% Confidence Interval of the Odds Ratio. McFadden Pseudo R-Square = 0.066. Dependent variable categories, 1 = selected 'no grain', 0 = did not select 'no grain'.
(DOCX)

**S3 Table. Multinomial logistic regression parameter estimates for the Demographics Model.** [1] Estimated multinomial logistic regression coefficient. [2] Odds Ratio or exponentiation of the coefficient (β). [3] 95% Confidence Interval of the Odds Ratio. McFadden Pseudo R-Square = 0.061. Dependent variable categories, 1 = selected 'no grain', 0 = did not select 'no grain'.
(DOCX)

**S4 Table. Multinomial logistic regression parameter estimates for the Exercise/Feeding Model.** [1] Estimated multinomial logistic regression coefficient. [2] Odds Ratio or exponentiation of the coefficient (β). [3] 95% Confidence Interval of the Odds Ratio. McFadden Pseudo R-Square = 0.060. Dependent variable categories, 1 = selected 'no grain', 0 = did not select 'no grain'.
(DOCX)

**S1 File.**
(PDF)

## Acknowledgments

We thank Dr. Chris Marinangeli (Pulse Canada) and Dr. Adronie Verbrugghe (Ontario Veterinary College) for their comments on the manuscript.

## Author Contributions

**Conceptualization:** Sydney Banton, Júlia G. Pezzali, Michael von Massow, Anna K. Shoveller.

**Data curation:** Sydney Banton, Andrew Baynham, Júlia G. Pezzali.

**Formal analysis:** Sydney Banton, Andrew Baynham, Michael von Massow.

**Funding acquisition:** Anna K. Shoveller.

**Investigation:** Sydney Banton, Andrew Baynham, Júlia G. Pezzali, Michael von Massow, Anna K. Shoveller.

**Methodology:** Sydney Banton, Andrew Baynham, Júlia G. Pezzali, Michael von Massow, Anna K. Shoveller.

**Project administration:** Sydney Banton.

**Supervision:** Michael von Massow, Anna K. Shoveller.

**Visualization:** Sydney Banton, Andrew Baynham, Michael von Massow.

**Writing – original draft:** Sydney Banton.

**Writing – review & editing:** Andrew Baynham, Júlia G. Pezzali, Michael von Massow, Anna K. Shoveller.

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
