## [Decision Letter · Decision Letter 0]

29 Jan 2021

PONE-D-20-39049

Grains on the brain: A survey of dog owner purchasing habits related to grain-free diets

PLOS ONE

Dear Dr. Shoveller,

Thank you for submitting your manuscript to PLOS ONE. After careful consideration, we feel that it has merit but does not fully meet PLOS ONE’s publication criteria as it currently stands. Therefore, we invite you to submit a revised version of the manuscript that addresses the points raised during the review process.

We look forward to receiving your revised manuscript.

Kind regards,

Juan J Loor

Academic Editor

PLOS ONE

Journal Requirements:

2.Thank you for stating the following financial disclosure:

 "This research was funded by Rolf C. Hagen, Inc. (Grant 053974, website: http://ca-en.hagen.com) and awarded to AKS. Warren House and Tiana Owens, employees of Rolf C. Hagen, Inc. both reviewed the manuscript."

3.Thank you for stating the following in the Competing Interests section:

"I have read the journal's policy and the authors of this manuscript have the following competing interests: Sydney Banton was funded by Rolf C. Hagen, Inc. to complete the research. "

Reviewers' comments:

Reviewer's Responses to Questions

**Comments to the Author**

1. Is the manuscript technically sound, and do the data support the conclusions?

Reviewer #1: Yes

Reviewer #2: Partly

Reviewer #3: No

2. Has the statistical analysis been performed appropriately and rigorously? 

Reviewer #1: Yes

Reviewer #2: Yes

Reviewer #3: Yes

3. Have the authors made all data underlying the findings in their manuscript fully available?

Reviewer #1: Yes

Reviewer #2: Yes

Reviewer #3: Yes

4. Is the manuscript presented in an intelligible fashion and written in standard English?

Reviewer #1: Yes

Reviewer #2: Yes

Reviewer #3: Yes

5. Review Comments to the Author

Reviewer #1: PONE-D-20-39049 Grains on the brain: A survey of dog owner purchasing habits related to grain-free diets

I found the manuscript interesting and well organized. I think the work has value and will be useful to many in the pet food area. My only concern is that the paper is quite long; however, I have no concrete suggestions for shortening. My comments were quite limited.

Line

348 ‘about pet food from a various sources and’ awkward? Perhaps delete ‘a’

382 Suggest ‘or the USA’

466 Suggest ‘from the UK.’

678 ‘pet foods with the claims ‘no additives/preservatives’ and ‘all natural product’ were 4 and 7 times higher’ Suggest ‘the number of pet foods…’

Reviewer #2: The authors have described the results of dog owner survey focused on feeding habits of people living in a few countries in Europe and North America. The topic is of interest and a lot of data were generated, but feel major revisions are needed. Comments are provided below.

General comments:

1. The title and introduction are focused on grain-free diets (most likely to drive reader interest), but much of the data have nothing to do with this topic. I think a much more appropriate and useful approach is the focus on differing responses across countries, owner sex, etc. and to point out how much pet owners rely on fad diet, trends, and marketing strategies – many that have no scientific basis – to make their purchases. This is interesting, but a huge problem that the industry has. This dataset shows a lot of these problems in the human and pet food industries. While the authors report the data obtained, they have not tried too hard to point out that the lack of education, science, and facts that contribute to these choices is big problem. Only 1-2 sentences in the discussion highlight this problem (lines 625-628; lines 638-643). I would encourage the authors to take on that challenge.

2. If the authors feel they need to focus on grain-free, the first 7-8 figures should be moved to supplementary files. If that is the topic, it just distracts from the main point of the paper. It provides a foundation and background of the survey participants, but most of that data has nothing to do with that topic. The text can refer to the figures, but provide that information.

3. The manuscript is much too long. If data are provided in tables and figures, every data point is not necessary in the text of the Results section. The main differences should be provided in the text with reference to figures and tables.

4. The manuscript contains too much slang. Phrases like we found, pet owners get, etc. should be replaced with technical terms.

5. It is stated that because an independent firm provided the survey that it was unbiased. However, there is no evidence to back up this comment. How were people recruited? What groups were targeted? Unless this can be described, this statement is incorrect. Also, a huge bias is that only kibble feeders completed the survey. I think much of the data provided would be quite different for those feeding other dietary formats. This needs to be highlighted more prominently in the abstract, limitations section and conclusion.

6. In many countries, the consumption of whole grains is promoted as part of a healthy diet in humans. The benefits of consuming whole grains, which is science-based, is important information that should be highlighted and referenced. The disconnect on facts like this between human and pet nutrition is confusing and frustrating. If the authors do not provide science-based information like this, the data may actually feed the negative, ingredient-based trends that we are seeing in the industry today.

7. Figures must be reordered so that they are in order as presented. Figure 3 is the main problem.

Specific comments:

1. Title is inaccurate as written. It is suggested to remove the emphasis on grain-free, as it is only part of the dataset. If the focus remains on grain-free, than much of the results should be minimized and moved to supplemental files because it distracts from this point.

2. Abstract: the fact that the survey was only completed by kibble feeders is an important point that should be included. The responses may be drastically different for those feeding other diet formats.

3. Line 54-67 and 701-704: is the “natural” and “organic” trend really linked with the “grain-free” trend? The natural and organic trends are based on how plants are grown, how animals are fed and raised, and how ingredients are processed regardless of type. Grain-free is completely based on ingredient source and type.

4. Line 125 and other places: data are plural so replace “was” with “were”

5. Results section should be shortened dramatically. Exact data numbers are not needed in the figures, tables, and text. The text should provide the main differences and referring to tables and figures – not repeating them.

6. Line 196, 199 and other places: use “or” instead of “and”

7. Line 206-218: shorten significantly. Many of the specific points (lines 211-213) are inconsequential to the big picture and can be removed.

8. Line 272-292: shorten significantly.

9. Line 368: how are protein ingredients defined here? Many ingredients contain protein and AA but are not considered protein-rich ingredients and are not included for that purpose.

10. Discussion: can shorten by one third.

11. Line 553-554: see comments about bias above. Nearly everything has bias. It is not a problem for this study, but is important to highlight what they are and take them into account when interpreting the data.

12. 566-567: need to revise. Allergies to grain are based on their protein content. What are you meaning to say here?

13. Line 588-590: see comments above about this. Why does science demonstrate that whole grain consumption in humans is positive and part of the healthy dietary guidelines provided by governmental agencies, yet grains are perceived to be a problem with pet owners? This is a very important disconnect that needs to be addressed.

14. Line 595-597: this suggests that this group is the “fad diet” group, a group that probably tries a lot of strategies based on marketing rather than the boring, science-based strategy of eating a balanced diet, regular exercise, and low quantities of negative behaviors (drinking, smoking). This problem should be called out.

15. Line 603: the term “since” refers to time. It should be replaced with a more appropriate term.

16. Line 611-614: This again highlights one of the major problems with pet food consumers today. Completely stuck on ingredients rather than scientific facts. I would encourage the authors to not only present these data, but discuss why that type of thinking is problematic.

17. Line 625-628 and line 638-643: important points that deserve more attention.

18. Line 643-654: delete. This supposed problem has been ingredient based from the beginning and lacking basic science principles. This section continues this problematic narrative.

19. Line 684-686: what do French human dietary recommendations say about grains? This information is necessary in order to properly interpret these responses.

20. Line 703: natural and grain-free trends are quite different. Why are the authors trying to link them together?

21. Line 709-716: a stronger conclusion is needed, stressing the problems with ingredient-based marketing and purchasing trends. If researchers don’t highlight these problems, who will? Also, this dataset is completely based on opinions of kibble diet feeders. Data from those feeding other formats may be quite different.

22. Figure 7: spell out veterinarian

Reviewer #3: The manuscript reports a survey based on the classification of grain free food for companion animals. The criticism is that the term "grain free" is not defined for the owners and has not scinetific basis, is only a marketing matter. In other words, the report is useful for a feed company and for marketing purpose but does not add anything for the scientific community.

6. PLOS authors have the option to publish the peer review history of their article (what does this mean?). If published, this will include your full peer review and any attached files.

Reviewer #1: No

Reviewer #2: No

Reviewer #3: No

---

## [Author Response · Author response to Decision Letter 0]

22 Feb 2021

Response to reviewer #1

Thank you for the comprehensive review. We believe that addressing your queries significantly improved our manuscript. We specifically worked to shorten the results and discussion section. We also made the suggested word or phrase changes outlined in your feedback. 

Response to reviewer #2

General comments:

1. The title and introduction are focused on grain-free diets (most likely to drive reader interest), but much of the data have nothing to do with this topic. I think a much more appropriate and useful approach is the focus on differing responses across countries, owner sex, etc. and to point out how much pet owners rely on fad diet, trends, and marketing strategies – many that have no scientific basis – to make their purchases. This is interesting, but a huge problem that the industry has. This dataset shows a lot of these problems in the human and pet food industries. While the authors report the data obtained, they have not tried too hard to point out that the lack of education, science, and facts that contribute to these choices is big problem. Only 1-2 sentences in the discussion highlight this problem (lines 625-628; lines 638-643). I would encourage the authors to take on that challenge.

2. If the authors feel they need to focus on grain-free, the first 7-8 figures should be moved to supplementary files. If that is the topic, it just distracts from the main point of the paper. It provides a foundation and background of the survey participants, but most of that data has nothing to do with that topic. The text can refer to the figures, but provide that information.

3. The manuscript is much too long. If data are provided in tables and figures, every data point is not necessary in the text of the Results section. The main differences should be provided in the text with reference to figures and tables.

Response to 1-3:

Thank you for your thoughtful comments. We understand your point of view regarding fad diets, trends and differing approaches across countries; however, our a priori hypothesis and intent was directly related to grain-free diets due to the concern in North America that feeding these diets is associated with dilated cardiomyopathy. Furthermore, we used the consumer data in the multinomial models where many factors are associated with the choice of a grain-free diet, including the owner’s own dietary regimen, perception of allergies in dogs and even demographic variables such as sex and country. Consequently, it is necessary to report the data that focus on the basic understanding of the population as they are the foundations behind the statistical models. Our hypothesis states that we suspect those with more strict dietary regimens would be more inclined to select ‘no grain’ and that is what we found. Consumer research dictates that it is necessary to present the initial country differences in the variables that were used in the 3 models so that the reader can understand the differences between countries before the variables are incorporated into the models. In addition, we have added a line to the introduction to make this more clear: “In order to understand differences in consumer attitudes towards grain-free dog food, it is necessary to first understand differences in consumer demographics, diet and purchasing habits that may contribute to that.” (Line 71-74 in new manuscript). 

However, we understand that presenting all data may divert the focus of the manuscript, and thus, we have significantly decreased the data in the results section that is unrelated to the three models presented (mainly feeding frequency, dog’s body weight and dog’s exercise) and removed two figures (feeding frequency and dog’s exercise) that were unrelated to the variables in the models. We have also removed some of the text in the Results and referred to the figures instead as well as removed some of the numbers in the text that are already highlighted in the tables (specifically the ORs and CI in the regressions).

4. The manuscript contains too much slang. Phrases like we found, pet owners get, etc. should be replaced with technical terms.

We have made phrases like this more formal. For example, line 40 changed from “this survey gives us insight” to “this survey provides insight,” line 497 removed “we found that.” 

5. It is stated that because an independent firm provided the survey that it was unbiased. However, there is no evidence to back up this comment. How were people recruited? What groups were targeted? Unless this can be described, this statement is incorrect. Also, a huge bias is that only kibble feeders completed the survey. I think much of the data provided would be quite different for those feeding other dietary formats. This needs to be highlighted more prominently in the abstract, limitations section and conclusion.

We did not state that the survey was completely unbiased, we highlighted the fact that a third party did all of the recruitment via email which does decrease the location/distribution bias. Most of the published manuscripts that use surveys to understand any aspect regarding the pet food industry and/or pet food owners use social media platforms to recruit participants (Dodd et al., 2019, Schleicher et al., 2019, Morelli et al., 2019), which does not represent the general population. Qualtrics is a recognized survey software that is widely used in other areas of research to understand consumer habits. 

We do appreciate your comment about this being biased towards pet owners who choose dry dog food. Seeing as we excluded respondents who feed other dietary formats, we have added “grain-free dry dog food” to the title and made an effort to point out more often in the text that these were respondents who feed dry dog food only.

6. In many countries, the consumption of whole grains is promoted as part of a healthy diet in humans. The benefits of consuming whole grains, which is science-based, is important information that should be highlighted and referenced. The disconnect on facts like this between human and pet nutrition is confusing and frustrating. If the authors do not provide science-based information like this, the data may actually feed the negative, ingredient-based trends that we are seeing in the industry today.

Thank you for addressing this, it is a great discussion point. We agree that this is an important disconnect between human and pet food. We have made an effort to address this in the discussion section, Line 529-536 in new manuscript:

“The above conclusions highlight an important disconnect between human and pet nutrition. While the majority of European and North American food-based dietary guidelines recommend consuming whole grains as part of a healthy diet [17], grains appear to be perceived as unhealthy in a dog’s diet by pet owners. There is ample evidence that consuming whole grains and fibre can lower human risk for cardiovascular disease [18] and no scientific evidence that grains are detrimental to a dog’s health. This suggests that the beliefs dog owners have about grains may be largely based on marketing strategies, which is consistent with the findings that selecting claims, ‘no fillers’ and ‘no by-products’ are associated with selecting ‘no grain.’”

7. Figures must be reordered so that they are in order as presented. Figure 3 is the main problem.

Thank you for pointing this error out. We have gone through the figures and renumbered them to ensure they are ordered correctly. 

Specific comments:

1. Title is inaccurate as written. It is suggested to remove the emphasis on grain-free, as it is only part of the dataset. If the focus remains on grain-free, than much of the results should be minimized and moved to supplemental files because it distracts from this point.

See our response to 1-3 above. Additionally we have added “dry dog food” to the title: “Grains on the brain: A survey of dog owner purchasing habits related to grain-free dry dog food.”

2. Abstract: the fact that the survey was only completed by kibble feeders is an important point that should be included. The responses may be drastically different for those feeding other diet formats.

Thank you for highlighting this, it is an important aspect of the current study. We have now added “dry dog food” to the title and pointed out more explicitly that we are investigating how respondents select a grain-free dry dog food in the abstract and throughout the text. Instead of saying “grain-free diets” we have changed it to “grain-free dry dog food” throughout the text.

3. Line 54-67 and 701-704: is the “natural” and “organic” trend really linked with the “grain-free” trend? The natural and organic trends are based on how plants are grown, how animals are fed and raised, and how ingredients are processed regardless of type. Grain-free is completely based on ingredient source and type.

Thank you for this comment, we agree. We have removed the sentence in the introduction where we classify grain-free as ‘natural’ pet food. Additionally, we have changed the first sentence of the abstract from, “Natural pet food options abound in the pet food market…” to “Grain-free pet food options abound in the pet food market today, representing more than 40% of available dry dog foods in the United States.” The paragraph that contains line 701-704 has been removed in order to decrease the length of the discussion as it was focused on a result from the supplementary models. We have changed the sentence comparing natural pet food market share between the US and UK to a comparison of grain-free market share: “In 2015, when grain-free was nearing its peak in the USA, almost 30% of market share was composed of grain-free pet food, compared to only 15% in the United Kingdom and 1% in France in the same year [5].” (Line 64-66 in new manuscript). Consequently reference number 5 was changed to a report from Growth from Knowledge on the pet food sector in 2016. 

It is also important to note, we never actually used or defined the term “grain-free” in the survey and instead used “no grain”. We did so intentionally to avoid any bias consumers may have when they hear the term “grain-free”. The term also carries some negative connotations in the USA after the FDA report so we intentionally did not define. The only term that appears as one of the options in the pet food questions is “no grain.” We have added this to the methods section:

“There were three separate questions that asked what features of pet food respondents look for when selecting a pet food. One of these questions contained ‘no’ statements, including ‘no grain.’ The term ‘no grain’ was intentionally used instead of the term ‘grain-free’ in order to avoid introducing any negative bias towards the term ‘grain-free’ due to the 2018 FDA report [8] suggesting a link between grain-free diets and the development of canine dilated cardiomyopathy.” (Line 103-108 in new manuscript)

4. Line 125 and other places: data are plural so replace “was” with “were”

Thank you, this has been corrected.

5. Results section should be shortened dramatically. Exact data numbers are not needed in the figures, tables, and text. The text should provide the main differences and referring to tables and figures – not repeating them.

We have removed significant portions of the results section, mainly results that are not pertinent to the models like dog’s body weight, feeding frequency and exercise as well as the two figures that were associated with these results. Additionally, we have removed all ORs and CIs from the text and referred to the table where the data in presented for all regression models.

6. Line 196, 199 and other places: use “or” instead of “and”

Thank you, this has been corrected.

7. Line 206-218: shorten significantly. Many of the specific points (lines 211-213) are inconsequential to the big picture and can be removed.

This has been shortened. The specific country comparisons have been removed and only the main distinction between France and the other countries has been called out in the text.

8. Line 272-292: shorten significantly.

This has been shortened. All of the data related to feeding frequency has been removed.

9. Line 368: how are protein ingredients defined here? Many ingredients contain protein and AA but are not considered protein-rich ingredients and are not included for that purpose.

We did not define the term protein in this context, the question in the survey simply stated, “When choosing a pet food, I look for… (select all that apply).” There were 3 of these questions in the survey, so we assigned a broad term to each one (targeted nutrition related claims, protein related claims and ‘no’ statements) for the purposes of the results section only. This was done to inform the reader of the types of attributes that were listed in each question. We have also attached the full survey as part of supplemental material if the reader wanted to refer to the question in the context of the survey. We have also changed our wording to “protein related claims” in Line 291 of the new manuscript. 

10. Discussion: can shorten by one third.

We have made an effort to shorten the discussion by removing the paragraph that was discussing results related to demographics because those results were presented in the supplemental material. The discussion is now only 6 pages.

11. Line 553-554: see comments about bias above. Nearly everything has bias. It is not a problem for this study, but is important to highlight what they are and take them into account when interpreting the data.

We agree that everything has some form of bias. In this statement, we were trying to point out a strength of this survey being recruited by a third party rather than by social media, therefore eliminating much of the recruitment bias. We have also made more of an effort to address the dry food related bias throughout the text and how these results may be different if people who feed other dietary formats are included. 

12. 566-567: need to revise. Allergies to grain are based on their protein content. What are you meaning to say here?

Thank you for pointing this out, we were referring to animal protein sources being more common allergens than grain sources and have added the words ‘animal protein’ to make this more clear. The new sentence reads: “While animal protein allergies are far more common than grain allergies in dogs [9, 10], the results from the current study suggest that consumers may believe that a grain-free diet can address allergy symptoms in their dog.” Line 469-471 in new manuscript. 

13. Line 588-590: see comments above about this. Why does science demonstrate that whole grain consumption in humans is positive and part of the healthy dietary guidelines provided by governmental agencies, yet grains are perceived to be a problem with pet owners? This is a very important disconnect that needs to be addressed.

Fantastic comment, thank you. This is a nice contrast between human and pet food that adds to the discussion. We have brought this point in and highlighted the disconnect between human and pet nutrition. We have also reworked this section of the discussion to address comments (16, 17 and 18) below as well. The following is the paragraph we added/re-worked:

“The above conclusions highlight an important disconnect between human and pet nutrition. While the majority of European and North American food-based dietary guidelines recommend consuming whole grains as part of a healthy diet [17], grains appear to be perceived as unhealthy in a dog’s diet by pet owners. There is ample evidence that consuming whole grains and fibre can lower human risk for cardiovascular disease [18] and no scientific evidence that grains are detrimental to a dog’s health. This suggests that the beliefs dog owners have about grains may be largely based on marketing strategies, which is consistent with the findings that selecting claims, ‘no fillers’ and ‘no by-products’ are associated with selecting ‘no grain.’ However, allowing consumers to drive ingredient composition of pet food could potentially lead to nutrient imbalances if amino acid balance is not considered [15, 19]. Recently, regular consumption of a grain-free diet has been associated with the development of canine dilated cardiomyopathy (DCM) [8]. Grain-free diets replace grain ingredients with pulse ingredients. Although high in protein, the amino acid profile of a pulse is different than that of a grain. Thus, careful consideration should be taken with regards to the amino acid composition and balance of the diet when grains are completely replaced by legumes and/or pulses. Furthermore, the Purchasing Habits Model found that those who rotate their dogs dry food to provide variety are less likely to select ‘no grain’, suggesting that grain-free diets may be selected as a static and regimented dietary routine. A recent clinical study by Kaplan et al. [20] revealed that in 24 Golden Retrievers diagnosed with DCM, the median length of time consuming the same grain-fee diet was 814 days, and the maximum, 3,558 days. In human nutrition, a diverse diet is encouraged to ensure breadth of nutrient intake to maintain health. It has been well established that those in developing countries, particularly women, who lack a diverse diet are more likely to develop micronutrient deficiencies [21]. Furthermore, using data from more than 1,000 French adults, Bianchi et al. [22] showed that adults with a higher diet diversity score were more likely to have a higher nutritional adequacy score. Although the importance of a diverse diet is established in human nutrition, there is a dearth of data with regards to the metabolic and health impact of a static feeding regimen in the domestic dog. Thus, future studies should investigate the effects of static feeding of a grain-free diet on micronutrient imbalances and its impact on the development of nutritionally-mediated DCM.” Line 529-557 in new manuscript. 

14. Line 595-597: this suggests that this group is the “fad diet” group, a group that probably tries a lot of strategies based on marketing rather than the boring, science-based strategy of eating a balanced diet, regular exercise, and low quantities of negative behaviors (drinking, smoking). This problem should be called out.

This is a great observation and could be the case. However, in the question where we asked about diet of the respondent, there were several options that would not be considered ‘fad diets,’ such as, Kosher, Halal, low sodium, celiac, diabetic. Therefore, this group of respondents who selected they are following 5 or more dietary regimens could be following a number of diets for medical or religious reasons. Because we did not ask about fad diets specifically, we believe it may be difficult to infer this. 

15. Line 603: the term “since” refers to time. It should be replaced with a more appropriate term.

This has been corrected.

16. Line 611-614: This again highlights one of the major problems with pet food consumers today. Completely stuck on ingredients rather than scientific facts. I would encourage the authors to not only present these data, but discuss why that type of thinking is problematic.

See #13 above and additional changes in this area of the discussion. Line 529-557 in new manuscript. 

17. Line 625-628 and line 638-643: important points that deserve more attention.

See #13 above and additional changes in this area of the discussion. Line 529-557 in new manuscript. 

18. Line 643-654: delete. This supposed problem has been ingredient based from the beginning and lacking basic science principles. This section continues this problematic narrative.

See #13 above and additional changes in this area of the discussion. Line 529-557 in new manuscript. 

19. Line 684-686: what do French human dietary recommendations say about grains? This information is necessary in order to properly interpret these responses.

This is a great observation and surprisingly to us, the French dietary guidelines also suggest consumption of whole grains with every meal so it is unclear why so many more French people would disagree with the statement, “I try to eat grains as part of a healthy diet.” We have added a line to address this (Line 578-581: Further to that idea, more respondents in France disagreed with the statement, “I try to eat grains as part of a healthy diet” compared to any other country, despite the food-based dietary guidelines in France recommending consumption of starchy foods, especially whole grains, at each meal [29].). 

20. Line 703: natural and grain-free trends are quite different. Why are the authors trying to link them together?

The paragraph that contains line 703 has been removed in order to decrease the length of the discussion as it was focused on a result from the supplementary models.

21. Line 709-716: a stronger conclusion is needed, stressing the problems with ingredient-based marketing and purchasing trends. If researchers don’t highlight these problems, who will? Also, this dataset is completely based on opinions of kibble diet feeders. Data from those feeding other formats may be quite different.

While we do understand your point about these problems, we believe we have addressed these in the discussion now. We have identified in the conclusion that consumers that value ingredients are more likely to select a grain-free diet and that this should be considered when the health and well-being of dogs consuming a grain-free diet are examined. In addition, we have added a line to address the value of consumer surveys in relation to knowledge transfer: “In addition, the data from the present survey lends insights into effective avenues for the scientific community to facilitate knowledge transfer between themselves and consumers.” Line 595-597 in the new manuscript. 

22. Figure 7: spell out veterinarian

This change has been made. In addition, we uploaded new figures with the letters used to assign significance changed to descending order. 

Response to Reviewer #3

The manuscript reports a survey based on the classification of grain free food for companion animals. The criticism is that the term "grain free" is not defined for the owners and has not scientific basis, is only a marketing matter. In other words, the report is useful for a feed company and for marketing purpose but does not add anything for the scientific community.

Defining grain-free and providing various levels of information on a scientific basis would have been an interesting addition to this study. The goal of the study was to understand who is purchasing grain-free, where they purchasing it, why they are purchasing it and where they are getting their information about pet food from. If we were to provide a definition of grain-free, it may bias some of the responses, especially if the information was in contrast to what respondents currently understood as being “grain-free”. While the definition itself is not defined within the survey, the term “grain-free” was never actually used in the survey either. We intentionally used the term ‘no grain’ instead so that we did not introduce any bias. This is because, the term “grain-free” has gained some negative attention lately due to its possible connection to dilated cardiomyopathy (DCM) in dogs. We have also added a line to the methods to point this out: 

“There were three separate questions that asked what features of pet food respondents look for when selecting a pet food. One of these questions contained ‘no’ statements, including ‘no grain.’ The term ‘no grain’ was intentionally used instead of the term ‘grain-free’ in order to avoid introducing any negative bias towards the term ‘grain-free’ due to the 2018 FDA report [8] suggesting a link between grain-free diets and the development of canine dilated cardiomyopathy.” (Line 103-108 in new manuscript)

The factors described within this paper, such as the demographic information and avenues which pet food purchasers get their information, dictates how types of pet foods are purchased. This has an advantage in lending insights and identifying effective avenues for the scientific community to facilitate knowledge outreach between themselves and the consumer. Having this understanding of consumer data is critical to the scientific community in understanding real world consumer applications.

---

## [Decision Letter · Decision Letter 1]

14 Apr 2021

Grains on the brain: A survey of dog owner purchasing habits related to grain-free dry dog foods

PONE-D-20-39049R1

Dear Dr. Shoveller,

We’re pleased to inform you that your manuscript has been judged scientifically suitable for publication and will be formally accepted for publication once it meets all outstanding technical requirements.

Kind regards,

Juan J Loor

Academic Editor

PLOS ONE

Additional Editor Comments (optional):

Reviewers' comments:

Reviewer's Responses to Questions

**Comments to the Author**

1. If the authors have adequately addressed your comments raised in a previous round of review and you feel that this manuscript is now acceptable for publication, you may indicate that here to bypass the “Comments to the Author” section, enter your conflict of interest statement in the “Confidential to Editor” section, and submit your "Accept" recommendation.

Reviewer #1: All comments have been addressed

Reviewer #3: (No Response)

2. Is the manuscript technically sound, and do the data support the conclusions?

Reviewer #1: Yes

Reviewer #3: No

3. Has the statistical analysis been performed appropriately and rigorously? 

Reviewer #1: Yes

Reviewer #3: Yes

4. Have the authors made all data underlying the findings in their manuscript fully available?

Reviewer #1: Yes

Reviewer #3: Yes

5. Is the manuscript presented in an intelligible fashion and written in standard English?

Reviewer #1: Yes

Reviewer #3: Yes

6. Review Comments to the Author

Reviewer #1: (No Response)

Reviewer #3: Dear Authors,

concerning my previous comments, the title and the manuscript still contain "grain free" term. I read the comments of the other two reviewers and your answers. The manuscript was improved, but the main critcism I raised was not resolved. The aim of the paper is market oriented and useless from a scientific point of view.

7. PLOS authors have the option to publish the peer review history of their article (what does this mean?). If published, this will include your full peer review and any attached files.

Reviewer #1: No

Reviewer #3: No

---

## [Editor Report · Acceptance letter]

22 Apr 2021

PONE-D-20-39049R1 

Grains on the brain: A survey of dog owner purchasing habits related to grain-free dry dog foods 

Dear Dr. Shoveller:

I'm pleased to inform you that your manuscript has been deemed suitable for publication in PLOS ONE. Congratulations! Your manuscript is now with our production department. 

Kind regards, 

on behalf of

Dr. Juan J Loor 

Academic Editor

PLOS ONE